# Snapshot multidimensional photography through active optical mapping

Jongchan Park[1,2,4], Xiaohua Feng[1,2,4], Rongguang Liang[3] & Liang Gao [1,2,4 ✉]

Multidimensional photography can capture optical fields beyond the capability of conventional image sensors that measure only two-dimensional (2D) spatial distribution of light. By mapping a high-dimensional datacube of incident light onto a 2D image sensor, multidimensional photography resolves the scene along with other information dimensions, such as wavelength and time. However, the application of current multidimensional imagers is fundamentally restricted by their static optical architectures and measurement schemes—the mapping relation between the light datacube voxels and image sensor pixels is fixed. To overcome this limitation, we propose tunable multidimensional photography through active optical mapping. A high-resolution spatial light modulator, referred to as an active optical mapper, permutes and maps the light datacube voxels onto sensor pixels in an arbitrary and programmed manner. The resultant system can readily adapt the acquisition scheme to the scene, thereby maximising the measurement flexibility. Through active optical mapping, we demonstrate our approach in two niche implementations: hyperspectral imaging and ultrafast imaging.

[1] Department of Electrical and Computer Engineering, University of Illinois at Urbana-Champaign, Urbana, IL 61801, USA. [2] Beckman Institute for Advanced Science and Technology, University of Illinois at Urbana-Champaign, Urbana, IL 61801, USA. [3] College of Optical Sciences, The University of Arizona, Tucson, AZ 85721, USA. [4] Present address: Department of Bioengineering, University of California, Los Angeles, CA 90095, USA. ✉ email: gaol@ucla.edu

The light rays, characterised by a plenoptic function P($x$, $y$, $z$, $\theta$, $\varphi$, $\lambda$, $t$) ($x$, $y$, $z$, spatial coordinates; $\theta$, $\varphi$, emittance angles; $\lambda$, wavelength; $t$, time), contains the rich information of the object imaged. Conventional digital photography measures only light irradiance on a two-dimensional (2D) lattice, throwing away much of the information content along other dimensions. The insufficient measurement is a result of the light detection pipeline of current image sensors, where the received photons are converted to electrons and then read out sequentially after exposure. While the colour and angular information of light are averaged in the photon–electron conversion, the fast temporal information is lost during the pixel exposure and readout. Breaking these limitations and measuring photon tags in parallel has been the holy grail of multidimensional photography over the past several decades.

The primary challenge for multidimensional photography is to enable the measurement with only evolutionary changes to the standard image sensors, where the pixels are typically arranged in the 2D format. To measure a high-dimensional light datacube with a 2D image sensor, a common strategy is to trade a spatial axis for light information of another dimension, such as spectrum and time. For instance, to acquire the spectral information, a hyperspectral line camera[1,2] disperses the light using a prism or grating along a spatial axis, measuring a spatio-spectral ($x$, $\lambda$) slice with a 2D detector array per camera readout. Likewise, to acquire the temporal information, an ultrafast streak camera[3,4] deflects the light using a sweeping voltage, mapping the time-of-arrival of photons to a spatial axis and outputting a spatio-temporal ($x$, $t$) slice per readout. Despite being able to provide a high resolution, the drawback of these imagers is their reliance on scanning—to capture a three-dimensional (3D) ($x$, $y$, $\lambda$) or ($x$, $y$, $t$) datacube, they must scan along the other spatial dimension ($y$). The scanning mechanism poses a strict requirement on the repeatability of the scene—the light datacube that is visible to the instrument must remain the same during scanning. This condition is often compromised by the motion of the object in hyperspectral imaging or the stochastic nature of the event in ultrafast imaging.

By contrast, snapshot multidimensional imagers capture light datacube voxels in parallel, leading to a significant improvement in light throughput[5,6]. Rather than simply exchanging a spatial axis for other light information, snapshot multidimensional imagers adopt more complicated methods to map light datacube voxels to a 2D detector array for simultaneous measurement. Depending on the shared conceptual thread, snapshot multidimensional imaging techniques are generally based on two distinct methods. One method, referred to as direct measurement, constructs a one-to-one mapping relation between light datacube voxels and camera pixels. A simple remapping of measured data allows the reconstruction of the light datacube. Despite being computationally efficient, the direct measurement faces a fundamental limitation on the information content acquired—the number of light datacube voxels cannot exceed the total number of camera pixels. Representative modalities using this method encompass image mapping spectrometry[7] and sequentially timed all-optical mapping photography[8]. The second method, referred to as compressed measurement, allows multiple mapped light datacube voxels occupying the same camera pixel, achieving a greater detector utilisation ratio[9–12]. However, because of data multiplexing at the detector, the recovery of the light datacube is computationally costly. Moreover, the applicable scenes must be sparse in a specific domain. Within this category, representative techniques include coded aperture snapshot spectral imaging[13,14], adaptive feature-specific spectral imaging[15], coded aperture compressive temporal imaging[16], programmable pixel compressive camera[17], coded exposure photography[18], and compressed ultrafast photography[19].

Seeing its vast application in both basic science[20] and engineering[16,21–24], snapshot multidimensional imaging has experienced remarkable growth. Nonetheless, to date, most multidimensional imaging devices have been designed to work in a passive manner—the mapping relations between the light datacube voxels and camera pixels are fixed. The photographer often faces the dilemma of choosing an appropriate modality for a given application, particularly when there is a lack of prior information about the scene to be imaged. The lack of tunability, therefore, limits the range of applicability of current multidimensional imagers in demanding tasks.

To address this unmet need, herein we developed a tunable snapshot multidimensional photography approach through active optical mapping. We employ a high-resolution liquid crystal spatial light modulator (SLM) as an active optical mapper to permute the high-dimensional datacube voxels and map them onto a 2D image sensor. The resultant method enables, for the first time, adaptive measurement of a high-dimensional light datacube with a low-dimensional detector array, allowing a seamless transition between multiple imaging modalities in a single device upon demand.

## Results

**Active optical mapper**. We enable tunable multidimensional imaging by using an active optical mapper, which can adjust the mapping relation between the incident and emanated light on demand (Fig. 1). A high-resolution reflective-type SLM transforms a high-dimensional light datacube—which contains information unresolvable by a conventional 2D image sensor—into a low-dimensional array. With programmed mapping relations, one can reconstruct the high-dimensional light datacube from the 2D measurement by solving the correspondent inverse problem[25].

Noteworthily, unlike previous passive multidimensional imagers[7,26,27], our active optical mapper can readily tune the mapping relation tailored for a given application. Also, it allows a flexible switch between direct and compressed measurement upon demands. For instance, by displaying a linear phase array on the SLM, our active optical mapper functions as an array of angled mirror facets, slicing the image into stripes and creating blank spaces in between. Dispersing the resultant light field in other dimensions such as spectrum or time along the spatial axis fills the created blank spaces and yields a one-to-one mapping between high-dimensional light datacube voxels and detector pixels. Because the light datacube information can be directly inferred using a lookup table given by the scalar coefficients, there is no resolution loss, and the computation cost is negligible. Such a mapping method is ideal for surveillance applications where the results of the measurement must be displayed in real-time to provide closed-loop feedback[28]. By contrast, by presenting a pseudo-randomly distributed phase pattern on the SLM, our optical mapper encodes the image with the pseudo-random binary amplitude pattern in the spatial domain. Dispersing the resultant light field in other dimensions leads to a many-to-one mapping, allowing multiple remapped light datacube voxels to be measured by the same detector pixel. Such a mapping relation constructs the basis of the compressed sensing where the acquisition of a light datacube requires much fewer detector pixels provided that the original scene is sparse in a specific domain. The compressed measurement, therefore, has a unique edge in acquiring large-sized light datacubes with economical cameras such as those employed in point-of-care imaging applications.

**Demultiplexing a high-dimensional datacube**. Our method employs an optical architecture which demultiplexes a high-dimensional datacube using an active optical mapper, reducing

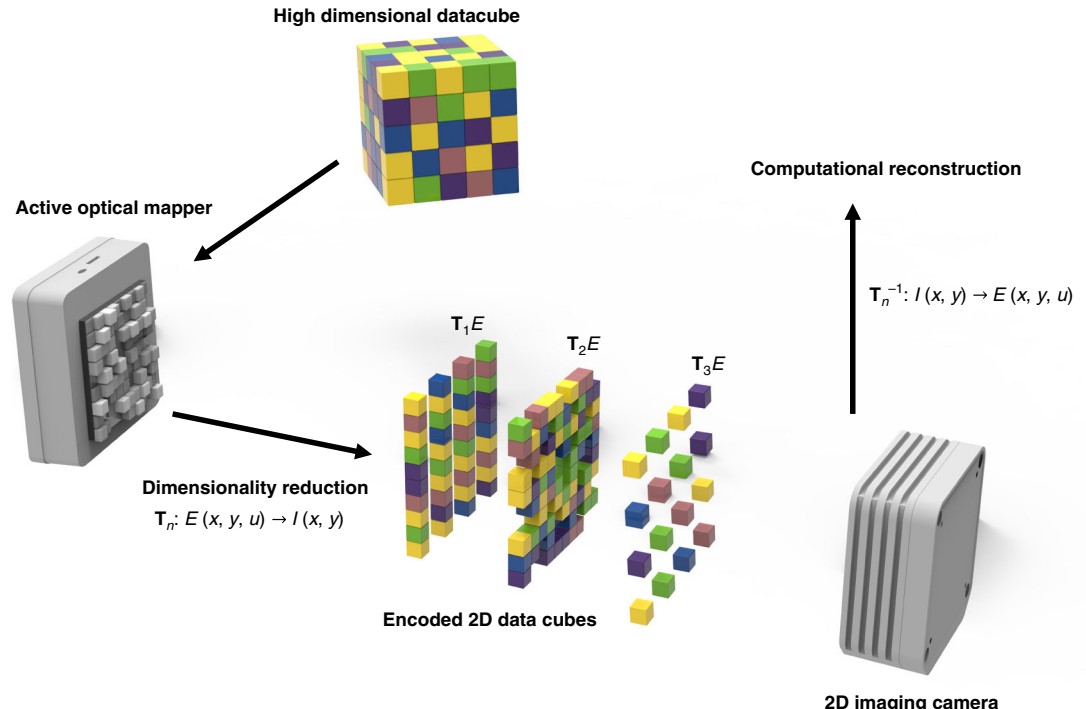

**Fig. 1 Operating principle.** A programmable mapper actively maps relations between a high-dimensional datacube of an incident light and a 2D image sensor on demand. With the mapping relations and the acquired 2D image data, the high-dimensional datacube can be computationally reconstructed.

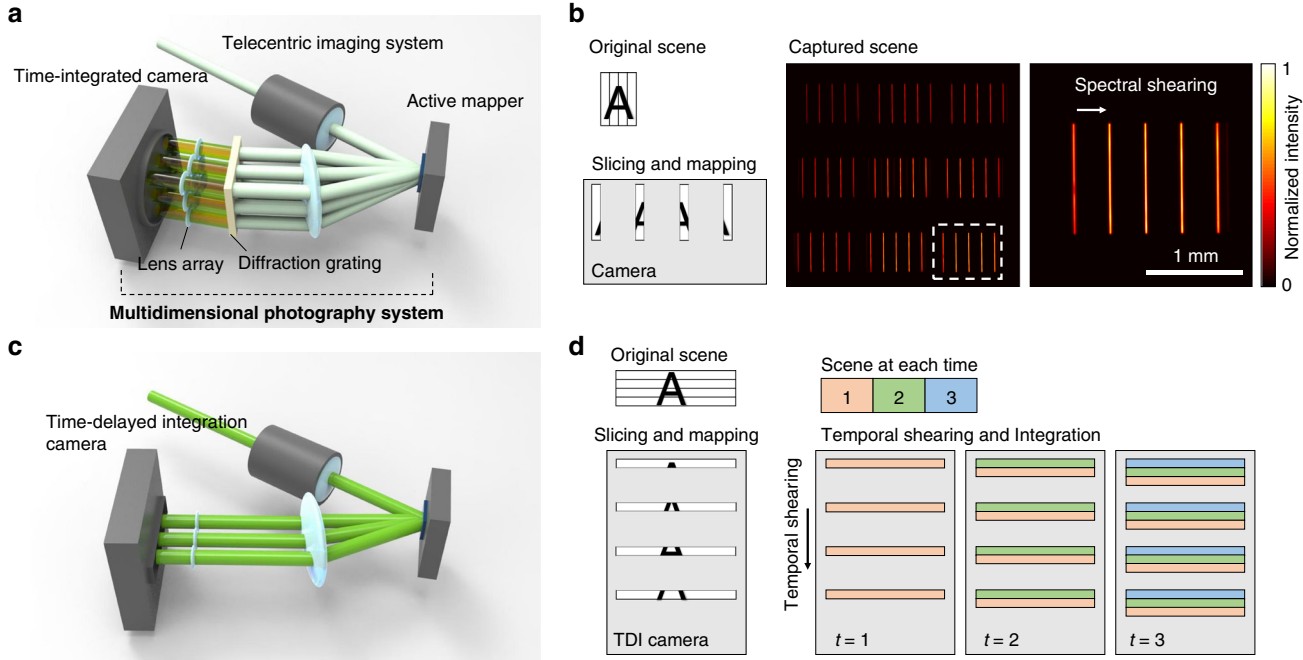

**Fig. 2 Demultiplexing a datacube along a specific dimension. a** Optical setup for $(x, y, \lambda)$ imaging. The optical setup consists of an active optical mapper, a diffraction grating, and multiscale imaging lenses. **b** The optical mapper vertically slices and redirects the incident image to reduce the dimensionality of the incident wave. The sliced image is dispersed horizontally by the diffraction grating and mapped onto a time-integrated 2D camera. **c** Optical setup for $(x, y, t)$ imaging. The optical mapper horizontally slices and redirects the incident image onto a time-delay integration (TDI) camera. **d** Operating principle of the TDI camera. TDI camera temporally shears the incident light signal along a spatial axis and integrates the signal along a temporal axis.

the information dimensionality and allowing the light datacube to be measured with a 2D image sensor. The schematic is shown in Fig. 2. The front-end optics generates an intermediate image and projects it onto an active optical mapper, the SLM, which imposes a programmed phase map on the incident light. Because the

spatial phase map of the light wavefront determines its propagation direction, the reflected light from the SLM is directed towards varied angles. The added phases to the image are thus transformed to the angles of outgoing rays, tearing apart the light datacube in the angular domain. To reimage the modulated light

field, we adopted a multiscale lens design[29], where a small-scale secondary lens array is placed at the back of a larger-scale objective lens. The angular separation created in the previous step is then converted to spatial separation at the Fourier plane of the larger-scale objective lens. Noteworthily, in our system, the SLM modulates the phase at a conjugate image plane, where the addition of the phase ramp to an image does not change the image intensity. Instead, it encodes the image segments with carrier spatial frequencies that correspond to the lateral displacements of small-scale lenslets on the array. Therefore, the incident light field is spatially rearranged, and the adjacent image segments in the original light datacube are separated and directed towards different small-scale lenslets, which further relay them to the 2D image sensor. To fill the spaces-created with information along other dimensions, we use dispersion elements such as a diffraction grating for hyperspectral imaging or time-delay integration (TDI) module for high-speed imaging. Such elements can be either inserted into the optical path or integrated into the detection process of the image sensor.

As an example, we implemented our acquisition scheme for hyperspectral imaging $(x, y, \lambda)$ and high-speed imaging $(x, y, t)$. For hyperspectral imaging (Fig. 2a, b), an incident 3D $(x, y, \lambda)$ light datacube is vertically sliced by the optical mapper, resulting in a set of 2D $(y, \lambda)$ images. By applying phase ramps to each sliced image, the images propagate to the desired directions. To resolve the spectral information, the sliced images are spectrally dispersed along $x$-direction by a diffraction grating and imaged onto a 2D detector through a lenslet array.

Likewise, our acquisition scheme enables measuring a 3D $(x, y, t)$ light datacube with a 2D image sensor (Fig. 2c, d). The resultant system allows recording of the transient dynamic scene with a frame rate beyond the data transfer bandwidth of conventional 2D image sensors. The incident 3D datacube is horizontally sliced by the optical mapper and temporally dispersed by a TDI camera[30,31] (Fig. 2c). In TDI operation, charge packets containing 2D image information are transferred along a spatial axis of the camera and continuously accumulate signals until being readout (Fig. 2d). Because the TDI operation is equivalent to temporally dispersing and integrating 2D image data into one-dimensional (1D) line data, the overall image-acquisition rate is given by the line-scanning rate of the TDI camera, which is several orders of magnitude faster than that of when operating the camera in the 2D-frame readout mode.

Our acquisition scheme also allows a smooth transition between direct and compressed measurements. Despite being computationally efficient, the one-to-one mapping between light datacube voxels and image sensor pixels limits the size of light datacube acquired at a single measurement. To measure large-sized light datacubes with a given image sensor, we introduced compressed sensing into our measurement scheme. The resultant compressed measurement is based on the same optical architecture of direct measurement. There is no mechanical alteration to the system during the switch. Instead, we changed only the phase pattern on the SLM to encode the light datacube with a pseudo-random binary pattern. The resultant spatially encoded light datacube is then dispersed along other dimensions (spectrum or time) and finally measured by a 2D image sensor. By allowing mapping of multiple voxels of the light datacube onto a single pixel of the sensor, compressed measurement significantly improves the measurement efficiency. Because the information is spatially multiplexed, a direct readout of the raw image provides no meaningful information as it is. Under sparsity constraints, the light datacube can be computationally reconstructed by solving the associated linear inverse problem by using optimisation algorithms such as a two-step iterative shrinkage thresholding algorithm[32].

**$(x, y, \lambda)$ imaging through direct optical mapping**. We demonstrated our system in hyperspectral imaging with direct optical mapping. The sample was a negative 1951 USAF target illuminated by two light beams of different colours (central wavelength, CWL = 532 nm/full width at half maximum, FWHM = 1 nm and CWL = 550 nm/FWHM = 10 nm) (Fig. 3a, b). By remapping the acquired 2D image data to 3D hyperspectral datacube voxels, we resolved two images with a significant spectral overlap, which the conventional colour camera fails to differentiate (Fig. 3c, d). A total of $45 \times 90 \times 18$ $(x, y, \lambda)$ voxels of the datacube were captured within a single snapshot. Noteworthily, we saved the mapping relation between the hyperspectral datacube and the 2D image sensor in a lookup table, thereby enabling real-time image reconstruction.

To further demonstrate our system in imaging real-world objects, we imaged an occluded finger (Fig. 3e). The tip of the forefinger was illuminated by a green light (CWL = 550 nm/ FWHM = 40 nm) and imaged by using our system. The hyperspectral datacubes were recorded at a video rate and reconstructed in real-time. During the measurement, the occlusion was released by cutting the rubber band. Immediately after the release of the occlusion, the total absorbance increased, which may result from reactive hyperaemia that the total haemoglobin concentration increases during the transient period of the increased blood flow (Fig. 3f). The absorbance spectrum of the occlusion-released finger is consistent with the molar attenuation coefficient spectrum of oxy-haemoglobin[33] where it shows a valley near 560 nm (Fig. 3h). After ten seconds, the total absorbance decreased as the blood flow rate, and the total haemoglobin concentration restored to the normal state (Fig. 3g).

**$(x, y, \lambda)$ imaging with compressive sensing**. To measure a large hyperspectral datacube, we switched the system to the compressed measurement mode. Noteworthily, the change of the acquisition scheme from direct to compressed measurement was accomplished simply by altering the phase pattern displayed on the active optical mapper. In practice, although the compressed sensing method can record and reconstruct the hyperspectral datacube beyond the Shannon–Nyquist sampling rate of the sensor, its reconstruction fidelity is primarily dictated by the sparsity of the signal and the sampling rate (or compression ratio). To mitigate this issue, we increased the sampling rate by using multiple encoding patterns—the input image was segmented and divided into nine groups in a pseudo-random manner owing to the flexibility of the optical mapper (Fig. 4a). The nine pattern-encoded hyperspectral datacubes were spectrally dispersed by the diffraction grating and imaged onto the camera at the different spatial positions through the lenslet array. Therefore, the total sampling rate was increased by a factor of nine compared to that of using standard encoding devices such as a single binary-coded aperture[13] or a digital micromirror device. This enhanced sampling rate enables the reconstruction of large-sized light datacubes with high fidelity.

To verify the capability of our system in acquiring large-sized hyperspectral datacubes, we captured a scene composed of beams passing through two 1951 USAF targets. One target was illuminated by a halogen lamp filtered by a 510 nm band-pass filter (CWL = 510 nm/FWHM = 10 nm). The other target was illuminated by both a narrow band 532 nm diode-pumped solid-state laser beam and a flashlight filtered by a 590 nm band-pass filter (CWL = 590 nm/FWHM = 10 nm). The two scenes were combined by a dichroic mirror and imaged by our system (Fig. 4b). The hyperspectral datacube was computationally reconstructed by solving the inverse problem associated with the programmed mapping relations (Fig. 4c). More frames are

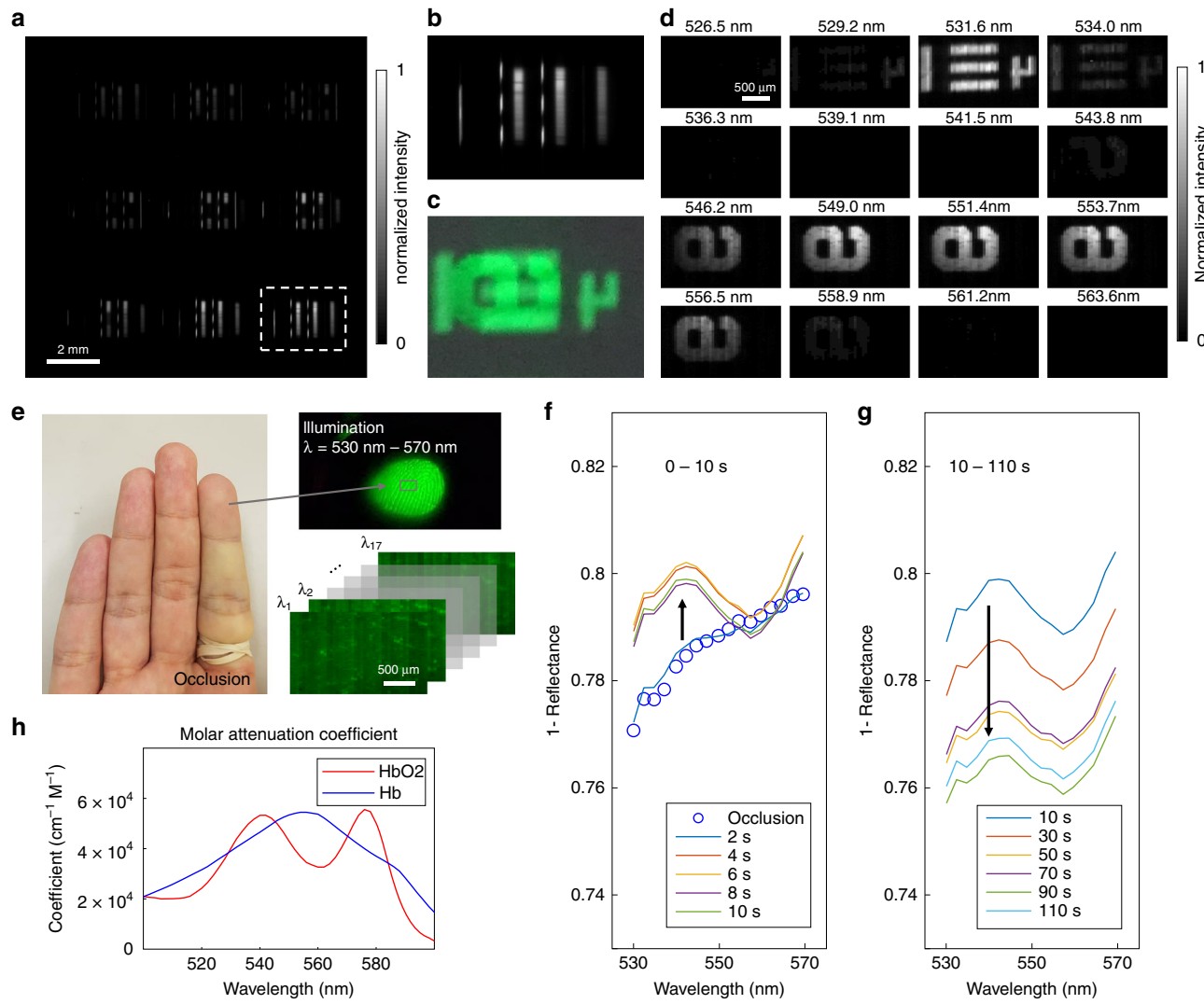

**Fig. 3 (x, y, λ) imaging through direct optical mapping. a** Grey-scale raw image captured by using multidimensional photography. **b** Magnified view of the image in (**a**). **c** Image captured by a commercial colour camera. **d** Hyperspectral images at each spectral channel. Two overlapped USAF target images are distinguished. **e** An artery in the forefinger was occluded with a rubber band. The tip of the finger was illuminated by a green light (CWL = 550 nm/FWHM = 40 nm). Spectral reflectance of the finger was captured after the removal of the rubber band. Short-term (**f**) and long-term (**g**) changes of the spectral response of the finger tissue. **h** Molar attenuation coefficient of oxy- and deoxy-haemoglobin[33].

provided in Supplementary Fig. 1 and Supplementary Movie 1. To assure sufficient sampling, we sampled a single resolvable optical mode (a voxel of the datacube) using approximately 3.5 camera pixels. The sampled size of the hyperspectral datacube was $400 \times 340 \times 275$ $(x, y, \lambda)$ voxels where the number of the resolvable spatial and spectral sampling modes was measured as $112 \times 96 \times 66$ $(x, y, \lambda)$. The spectral range was from 495 to 505 nm with 1.6 nm spectral resolution (Fig. 4d). The overall spectral irradiance matches well with the ground truth provided by a benchmark fibre spectrometer (STS-VIS-L-25-400-SMA, spectral resolution = 1.5 nm, Ocean Optics, Inc.).

**(x, y, t) imaging through direct optical mapping.** Next, we transformed our system to high-speed imaging. To demonstrate the measurement of a fast 3D $(x, y, t)$ scene beyond the frame rate of conventional 2D image sensors, we configured the system by replacing the original time-integration CCD camera in our previous hyperspectral imaging setup with a TDI camera. The role of the TDI camera is to temporally shear and integrate the incident light signal with high speed. This operation can be performed by

other temporal-shearing devices as well, such as a streak camera or a Galvano mirror[34] without losing generality.

We demonstrated direct optical mapping of a 3D $(x, y, t)$ transient event datacube onto the pixels of the 2D $(x, y)$ sensor. The imaging target was a fast-moving object (~4 m s$^{-1}$) illuminated by a laser beam ($\lambda = 532$ nm). The captured transient scene was horizontally sliced by the active optical mapper and imaged onto the TDI camera through the lenslet array (Fig. 5a). By remapping the 2D $(x, y)$ image captured by the TDI camera onto the 3D $(x, y, t)$ datacube voxels, we reconstructed a high-speed video of the fast-moving object. As shown in Fig. 5b, at 10.24 kHz frame rate, the motion blur is apparent that the image is elongated along the direction of the target's movement. At 153.6 kHz, the target image is clearly resolved given that the discrete artefact along the vertical direction is the result of the limited number of sliced images ($N = 27$). The full frames captured at 51.2 kHz are shown in Supplementary Fig. 2.

**(x, y, t) imaging with compressive sensing.** In the direct measurement mode, the maximum size of the time window at a single

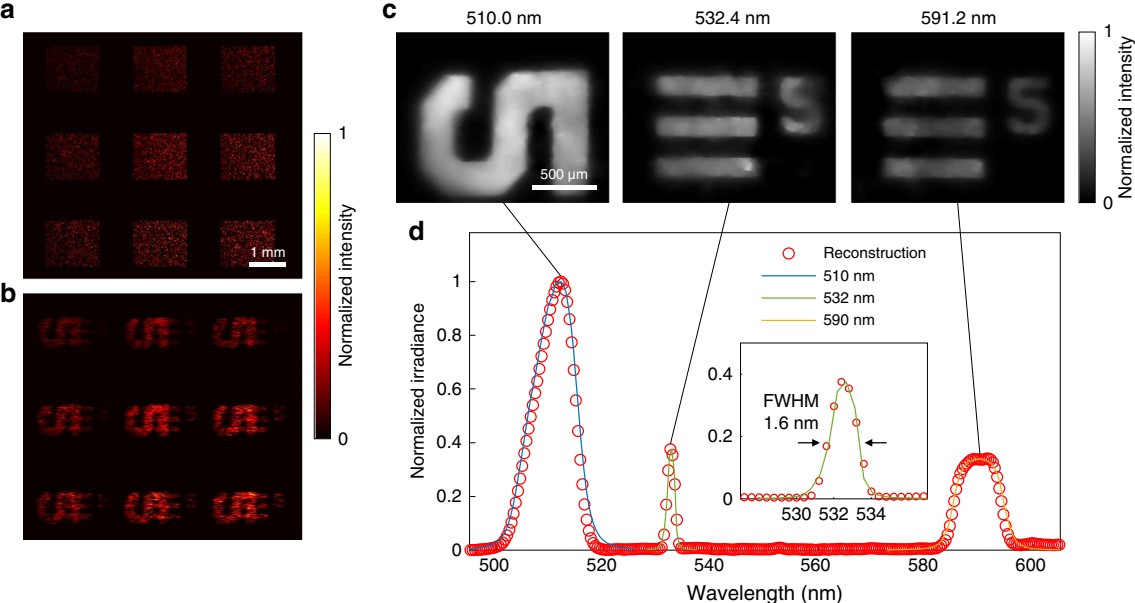

**Fig. 4 (x, y, λ) imaging through compressed optical mapping.** A hyperspectral datacube is spatially modulated by a coded aperture (**a**) and spectrally sheared by a dispersive element. **b** The spatially and spectrally encoded datacube is captured by a 2D image sensor. Computationally reconstructed hyperspectral images (**c**) and its averaged spectral irradiance (**d**).

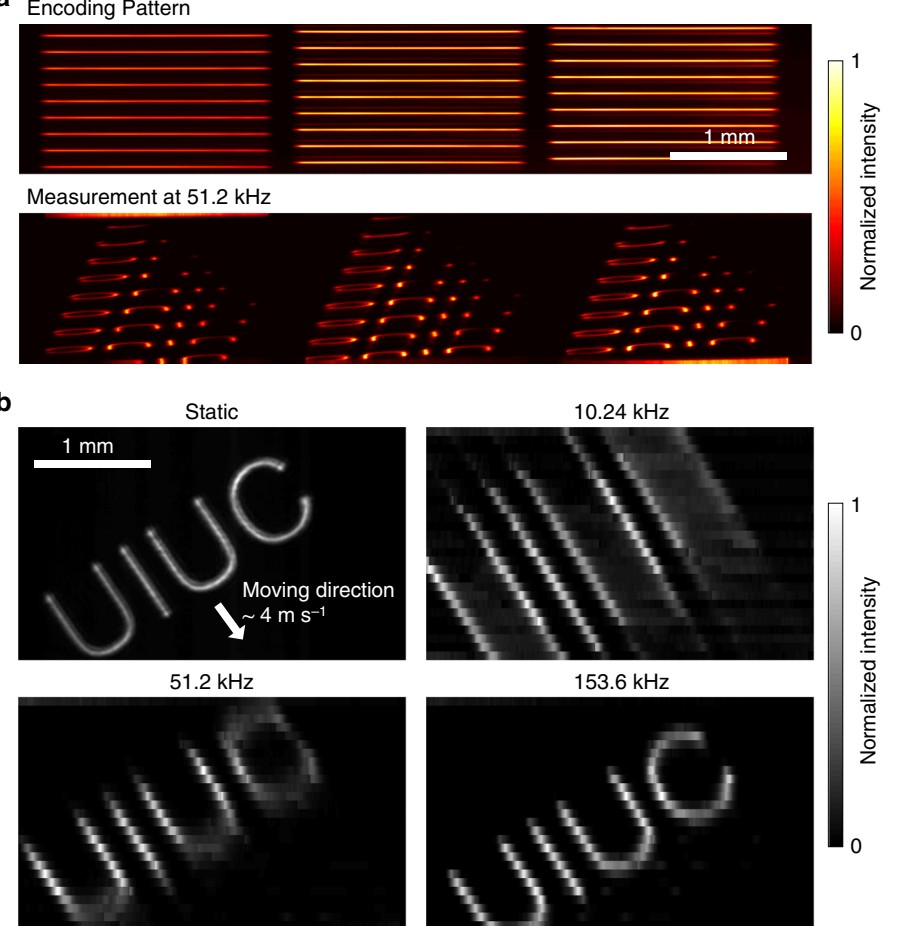

**Fig. 5 (x, y, t) imaging through direct optical mapping. a** An incident image is horizontally sliced by the active optical mapper and temporally sheared by the TDI camera. **b** Captured images at various frame rates.

measurement is given by the distance between the horizontally sliced images (pixels) divided by the TDI scanning rate (pixels per second). If the measurement time is longer than the designated time window, the temporally sheared scene is mapped onto the undesired positions of the sensor and the one-to-one mapping relation becomes invalid. Therefore, an optical chopper, a time-domain counterpart of the band-pass filter in the hyperspectral imaging, is required in order to confine the time window of the measurement.

To expand the time window, we switched the system to the compressed measurement mode. Rather than imposing a one-to-one relation between the incident 3D $(x, y, t)$ datacube and the sensor, the incident scene was fully recorded and integrated along the vertical direction with TDI operation allowing overlaps of the scene at different times. As the TDI camera integrates the photocurrent of 256 pixels along the vertical direction (256 TDI stages), the total compression ratio of the scene in the temporal

domain was 256. Specifically, the active optical mapper divided the incident image into three sub-images, each encoded by a complementary pseudorandom pattern. The encoded images then passed through the lenslet array, forming images at different spatial positions of the sensor in the TDI camera (Fig. 6b). The TDI camera temporally sheared, integrated, and streamed the incident scene at a line-scanning rate of 102.4 kHz (Fig. 6c) ("Methods"). After a computational reconstruction (Supplementary Movies 2 and 3), the fast-moving object, the rocket image, was resolved (Fig. 6d) at the given frame rate, 102.4 kHz.

We further demonstrated a niche application of our system in high-throughput imaging flow cytometry[35–37]. We flowed a fluorescent bead (diameter = 15 μm) in a custom microfluidic channel at a velocity of 0.8 m s$^{-1}$ using a syringe pump and illuminated the scene with a nanosecond pulsed laser ($\lambda = 532$ nm) (Fig. 6e). The emitted fluorescence was collected by an objective lens (×4/0.16NA), filtered by a dichroic mirror, and

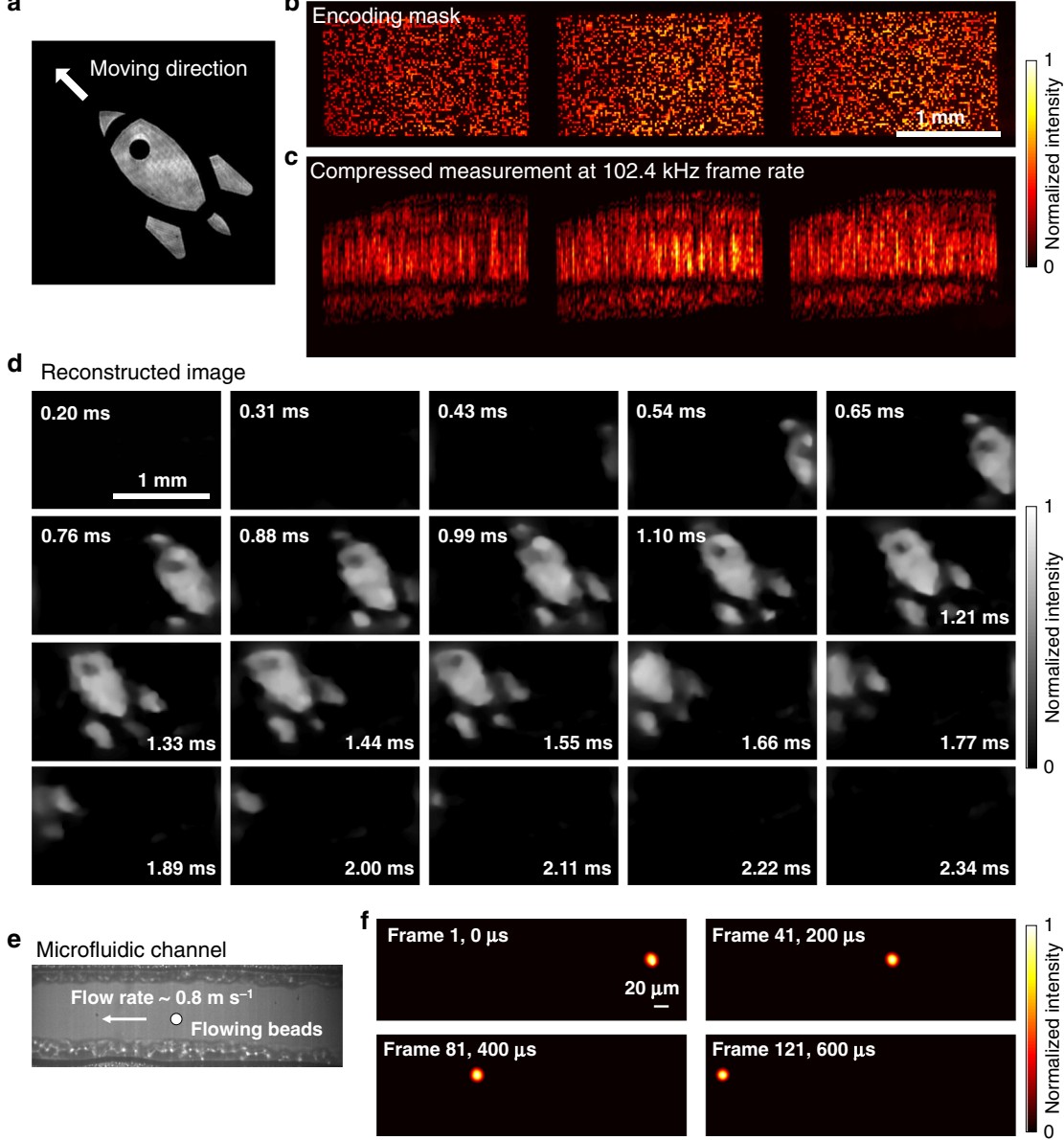

**Fig. 6 ($x, y, t$) imaging through compressed optical mapping. a** Target object. Three complementary masks (**b**) encode a dynamic scene. The scene was compressed and integrated into 1D line data by the TDI operation and continuously streamed to a host computer (**c**). **d** Computational reconstruction of the scene. The scene was captured at 102.4 kHz. **e** Photograph of a microfluidic channel. **f** Representative temporal frames of a flowing fluorescent bead imaged at 200 kHz.

imaged by our system at 200 kHz. The reconstructed blur-free fluorescent bead images at representative temporal frames are shown in Fig. 6f.

Noteworthily, our high-speed imaging scheme possesses several advantages over other state-of-the-art methods. To achieve a high frame rate, a conventional high-speed camera typically stores images on an internal memory chip[38] and transfers the data to a host computer after data acquisition. Therefore, the maximum time window per measurement is limited by the camera's on-chip memory size. By contrast, in our method, the TDI camera temporally shears and integrates 2D image data into a 1D line, thereby enabling a continuous stream of high-speed compressed images without imposing stringent requirements on the camera's data transfer rate. For instance, our high-speed imaging scheme can compress videos of $426 \times 256$ spatial pixels and continuously stream it at 200 kHz with 12 bit-depths. If the same scene is captured by a conventional 2D camera with a 32 GB on-chip memory, the maximum time window per measurement is 1.05 s. In addition, the cost and form factor of a TDI camera are much lower than those of a conventional high-speed camera with a similar frame rate.

**4D $(x, y, \lambda, t)$ imaging with hybrid optical mapping.** Finally, we demonstrated our system can perform four-dimensional (4D) $(x, y, \lambda, t)$ imaging by applying both the spectral and temporal shearing operations to the incident scene. The spectral and temporal shearing was accomplished by the diffraction grating and the TDI camera, respectively. By dispersing the spectral and temporal information along two orthogonal spatial axes, we reduced the dimensionality of the original 4D datacube to two, allowing it to be measured by a 2D image sensor in the snapshot mode.

Unlike a previous method which measures both spatial and spectral information in a compressed manner[39], we adopted a hybrid measurement scheme that blends direct and compressed measurements—while the time information is horizontally sliced and sheared along the vertical axis and directly mapped onto the image sensor within a given time window, the spectral information is sheared along the horizontal axis and measured by the compressed method. The rationale of using such a hybrid measurement scheme is that it can alleviate the trade-off between the measurable datacube size and image quality. On the one hand, in a direct measurement mode, a large number of detector pixels and SLM pixels is required to measure a 4D datacube with a high resolution. For instance, operating in the direct mapping mode, our system can acquire a 4D datacube of only $10 \times 10 \times 10 \times 10$ $(x, y, \lambda, t)$ voxels due to the limited pixel numbers of both the active optical mapper and the 2D sensor. On the other hand, although the compressed measurement can lead to a higher spatial, temporal, and spectral resolution, the reconstruction fidelity is highly sensitive to measurement noises. Our hybrid mapping method mitigates the above issues and enables high-quality measurement of $27 \times 54 \times 33 \times 10$ $(x, y, \lambda, t)$ 4D datacube in a single camera exposure.

To verify the snapshot 4D $(x, y, \lambda, t)$ imaging capability, we generated a 4D scene and imaged it using our system (Fig. 7a). A beam from a halogen lamp illuminated a linear variable filter, in which the spectral transmittance property varied continuously along the horizontal direction of the filter. Therefore, a rainbow-illumination beam was generated. A moving negative chrome mask was placed at the image plane of the linear variable filter. As a result, both the spectrally and temporally varying scene was encoded on the beam. The reconstructed 4D scene is shown in Fig. 7b, c. Figure 7b shows a spectrally integrated 2D $(x, y)$ scene at the given times. Figure 7c is a pseudo-coloured image that each colour stands for the maximum wavelength at the given

spatiotemporal point. Spectral irradiance at the given space and time is visualised in Fig. 7d. More illustrations are provided in Supplementary Fig. 3 and Supplementary Movie 4.

**System evaluation in direct measurement mode.** We quantitatively evaluated the system performance in the direct measurement mode. Without loss of generality, we assessed the system in spectral imaging of a standard colour checker target (X-Rite ColorChecker). We illuminated the colour checker with a green light (CWL = 550 nm, FWHM = 40 nm) and captured spectrum of the scattered light using both our system and a benchmark fibre spectrometer (O STS-VIS-L-25-400-SMA, Ocean Optics, Inc.) (Fig. 8). Here we use the fibre spectrometer to provide the ground-truth measurement. The root means squared errors (RMSE) of the normalised spectrum were quantitatively analysed (Fig. 8d). The average value of the RMSE was 0.11. Noises are mainly contributed by the stripe artefacts caused by nonuniform light diffraction efficiency of the SLM and unwanted diffraction orders of the beam. In addition, with a lower illumination irradiance, the RMSE tends to be higher because the signal-to-noise ratio (SNR) is relatively low.

**Direct measurement versus compressed measurement.** Our system presents a prominent advantage in flexible switching between direct and compressed measurements, both of which have pros and cons. On the one hand, while the direct mapping always yields accurate measurement due to the invertibility of unique optical mapping relation, it faces a trade-off between information content along myriad dimensions for a given detector array with a limited pixel number. On the other hand, the compressed measurement allows capturing a large-sized datacube beyond the Nyquist sampling rate of the system. However, the reconstruction process is computationally extensive, and it is prone to generate artefacts. For example, the reconstruction of a light datacube of size $400 \times 340 \times 275$ $(x, y, \lambda$ or $t)$ takes approximately twenty minutes on a personal computer (i7-8700 CPU, 6 cores, 3.2 GHz base clock rate).

To analyse at which conditions the compressed measurement scheme prevails over its direct counterpart, we evaluated the noise tolerance using a numerical approach. We generated a 3D ($150 \times 100 \times 50$) $(x, y, \lambda,$ or $t)$ datacube with varied SNR as the input scene, and we simulated three image formation models by encoding the scene with a binary pattern, three complementary patterns, and nine complementary patterns, respectively. The datacube was sheared and integrated along the third axis (spectrum or time). The resultant 2D scene was recorded by a 2D ($150 \times 100$) detector with white Gaussian noise corresponding to the given SNR. We defined the compression ratio, $\xi$, as the ratio of the voxel number of the incident 3D datacube to the pixel number of the 2D detector. For example, if the 3D ($150 \times 100 \times 50$) scene is encoded by a single binary pattern and one 2D ($150 \times 100$) image is captured, $\xi = 50$. If the scene is divided and captured through nine complementary encoding masks simultaneously ($150 \times 100 \times 9$), $\xi = 50/9$. If the 3D datacube is directly mapped onto a 2D sensor with a large number of pixels on a one-to-one basis, $\xi = 1$. The reconstruction results were quantified based on two metrics: the correlation coefficient and peak SNR. Figure 9 shows the reconstruction fidelity of the 3D datacube. When the SNR is 25 or higher, we can observe the expected scene in all simulations. However, the fine feature is visible only when the 3D datacube is captured with nine complementary encoding patterns ($\xi = 50/9$). In this case, the correlation coefficient between the noise-free 3D datacube reaches over 0.95, a level that is comparable to the direct mapping ($\xi = 1$) with the denoising operation[40]. Because the reconstruction fidelity highly

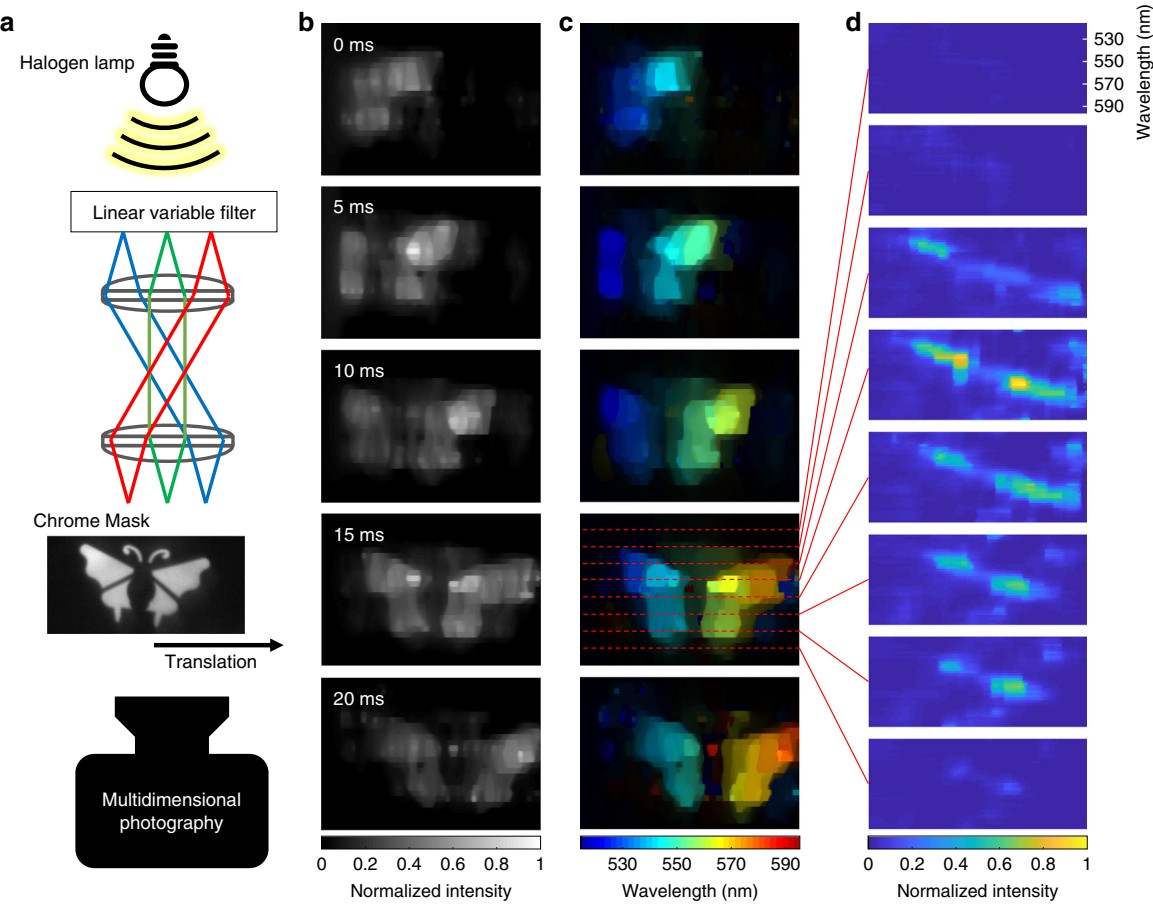

**Fig. 7 (x, y, λ, t) imaging with hybrid optical mapping. a** A beam pass through a linear variable filter illuminates a fast-moving mask to generate 4D scenes. Total irradiance (**b**) and pseudo-coloured image (**c**) of the captured 4D scene. The colour stands for the wavelength at the maximum irradiance at each spatial point. **d** Spectral irradiance of the scene at t = 15 ms.

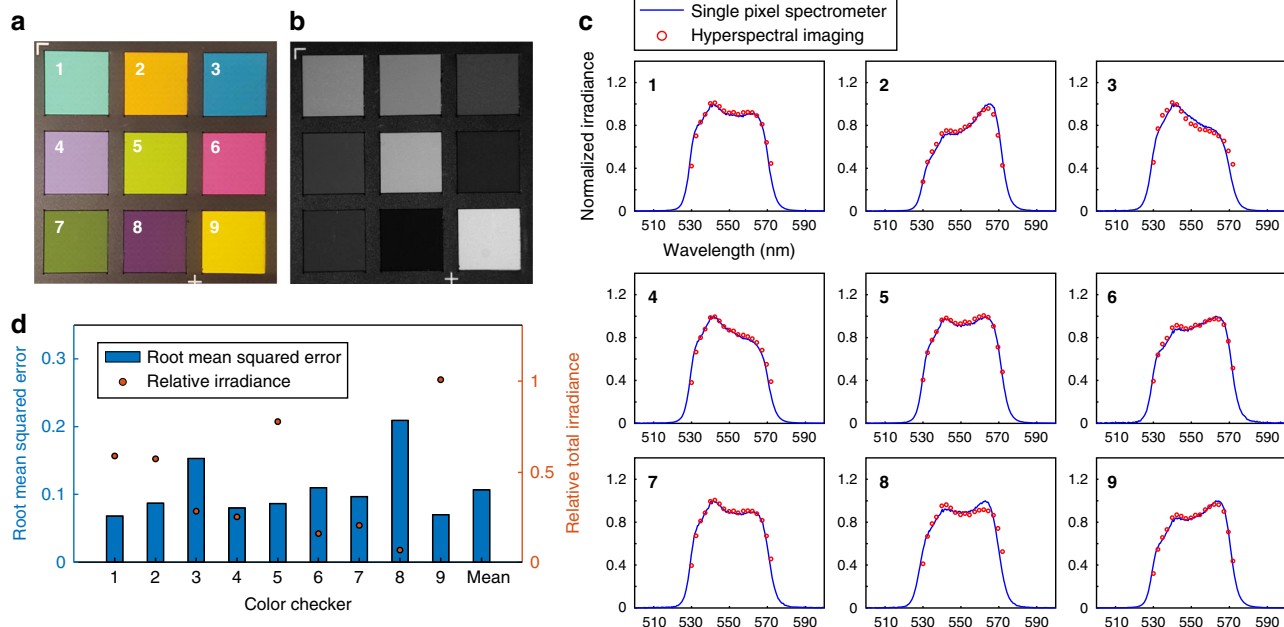

**Fig. 8 Technical evaluation of (x, y, λ) imaging with direct optical mapping. a** Photogram of the colour checker captured by a colour camera when it was illuminated by a fluorescent lamp. **b** Photogram of the colour checker captured by a monochromatic camera when it was illuminated by the green light (CWL = 550 nm, FWHM = 40 nm). **c** Normalised spectral irradiances of light scattered from the colour checker measured by multidimensional photography and a fibre spectrometer. The graphs indicate that our method can capture the hyperspectral datacube with high fidelity. **d** Root-mean-squared error (RMSE) of the hyperspectral images captured by multidimensional photography. The averaged RMSE value was 0.11.

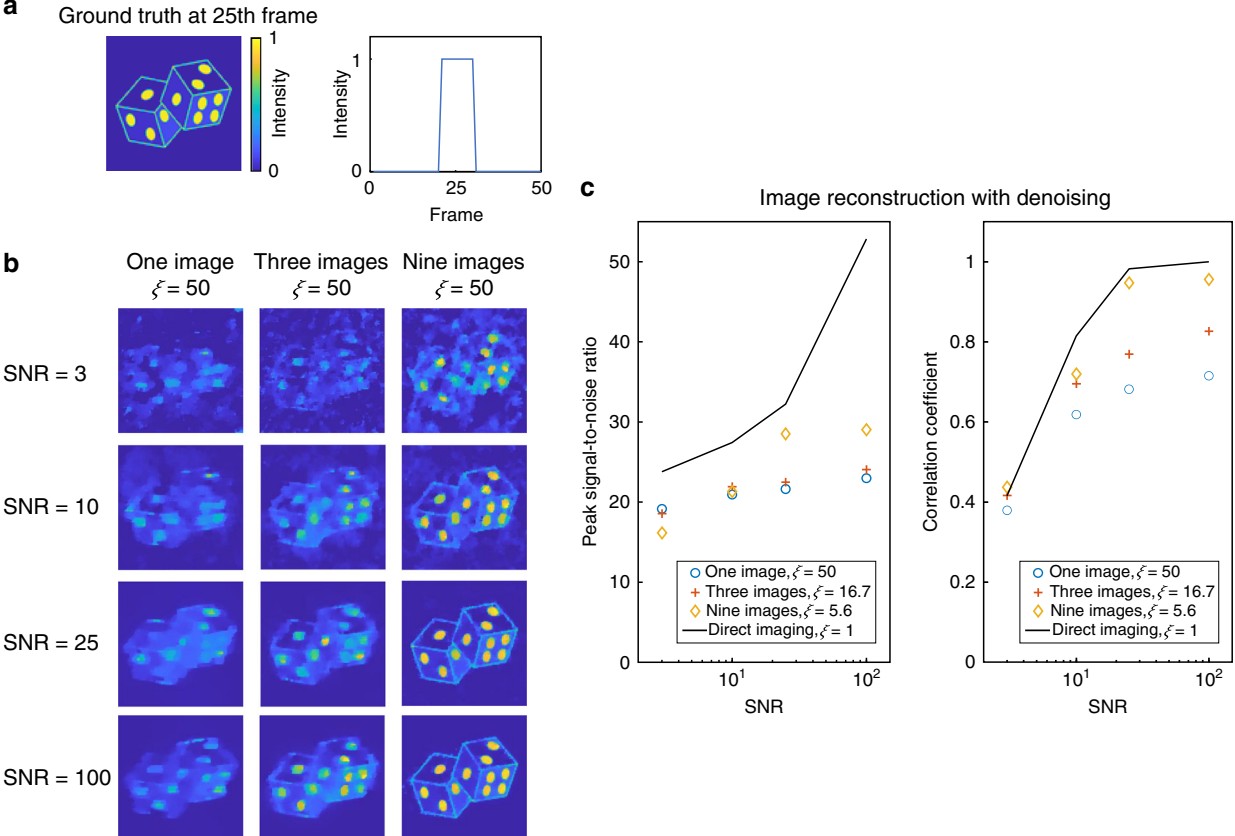

**Fig. 9 Effect of the compression ratio on compressed sensing. a** 3D datacube for the simulation. A total of 50 frames are compressed into a single frame. The 21st to 30th frames contain the image of the dice, and the rest 40 frames are blank. **b** Reconstructed 3D datacube. The 25th frames at given compression ratios and signal to noise ratios are shown. **c** Peak signal-to-noise ratio and correlation coefficient of the reconstructed datacube given the noise-free solution in (**a**).

depends on the sparsity of the signal, we cannot quantitatively compare the simulation with experiments. Nonetheless, the results herein exhibit the same trend as our previous experimental results—the hyperspectral imaging with nine complementary encoding patterns shows fine features (Fig. 4), while the ultrafast imaging with three complementary encoding patterns results in blurred edges (Fig. 6).

## Discussion

In conventional multidimensional optical imaging systems, the range of applicability of an image mapper is fixed upon being fabricated. For example, image spectrometers use passive image mappers, such as custom-designed mirror facets[7,41] or a bundle of fibres[27], to separate and redirect the incident image. Ideally, the spatial and spectral sampling rates are solely determined by the pixel resolution of the camera provided a full utilisation of camera pixels. However, it is challenging to fabricate a set of complex mirror facets with high surface quality that exactly meets the design requirement. Alternatively, using a fibre bundle offers flexibility on the optical design. Nonetheless, the small core area of the fibres limits the light transmittance.

By contrast, the active optical mapper can demultiplex high-dimensional datacubes on demand. For instance, in our hyperspectral imaging with direct optical mapping, instead of imposing only a set of spatial phase ramps (tilted plane mirrors) on the incident images, cylindrical or Fresnel lens patterns can also be applied. The addition of such phase maps on the incident image reduces the width of the sliced image in the dispersion axis,

thereby significantly increasing the spectral resolution of the system (Supplementary Note 1). The increase in the spectral resolution is at the expense of a reduced spatial resolution—the wider the original image segment, the greater the enhancement in spectral resolution, but the lower the spatial resolution. By contrast, in compressed measurement, both the spatial and spectral sampling rates can be enhanced simultaneously by reducing the minimum feature size of the encoding pattern. Therefore, there is no stringent upper bound on the total sampling rate. However, the computational complexity increases with the sampling rates.

In direct measurement, the product of the spatial and spectral/temporal sampling rates (i.e., the total number of voxels in the light datacube) is fundamentally limited by: (1) the number of controllable degrees of freedom of the pixelated SLM, which is referred to as the space-bandwidth product (SBP) of the system (Supplementary Note 2)[42], and (2) the total number of pixels of the image sensor. Because the pixel counts of large-format image sensors far exceed that of current state-of-the-art SLMs, the performance of our system is bounded by the SBP of the SLM. Yet, the use of a large-format image sensor or even an array of image sensors[29,43,44] is beneficial to image reconstruction, especially for a highly compressed measurement scheme[11].

Noteworthily, our active-mapping multidimensional photography employs a snapshot acquisition scheme, a fact that significantly improves the light throughput compared to the scanning-based methods, such as pushbroom and confocal scanners. Here, we define the light throughput as the percentage of light datacube voxels that are visible to the instrument at a single measurement[5]. For example, to measure a hyperspectral

datacube of a size $(x, y, \lambda) = (50, 50, 20)$, point-scanning and line-scanning systems require 2500 and 50 measurements, respectively. The corresponding light throughputs are 0.0004 and 0.02, respectively, wasting majority of light. In contrast, our system has a throughput close to one albeit a finite diffraction efficiency of the SLM.

In the current implementation, we used an SLM as an active optical mapper for snapshot measurement. Although such a device provides great flexibility in manipulating the angles of the reflected light, it poses a constraint on the total number of measurable light datacube voxels due to the SLM's limited space bandwidth product (Supplementary Note 2). This current limitation will be mitigated with continuing efforts of developing ultrahigh-resolution SLMs.

In conclusion, we developed and experimentally demonstrated a versatile snapshot multidimensional photography platform through active optical mapping. By exploiting the large degrees of freedom enabled by a high-resolution SLM, we can map voxels of a high-dimensional light datacube to a 2D image sensor in an arbitrary and programmed manner. The system can perform various tasks ranging from hyperspectral to ultrafast imaging in the single device upon demand.

## Methods

**Experimental system**. We collected a high-dimensional datacube by using a bi-telecentric lens (MVTC23100, Thorlabs, Inc.) and imaged it onto an active optical mapper through a 4-$f$ imaging system with a magnification of ×5.33. We used a reflective-type phase-only SLM (HSP1920-488-800-HSP8, Meadowlark Optics, Inc.) with a resolution of 1920 × 1152 pixels as the active optical mapper. A linear polarizer (LPVISE100-A, Thorlabs, Inc.) was placed before the SLM to enable phase-only modulation. By imposing a spatial phase map on the incident wavefront, the optical mapper modulated the propagating direction of the beam in a programmed manner. The reflected beams passed through an achromatic lens ($f = 250$ mm, diameter = 50 mm) and were directed to a diffraction grating (GT25-03, 300 grooves/mm, Thorlabs, Inc.) placed at the Fourier plane of the image. A custom-designed 3 × 3 lenslet array ($f = 30$ mm, diameter = 2.5 mm each) reimaged the spectrally sheared beam to a sensor. We used a monochromatic camera (3376 × 2704 resolution, 3.69 μm pixel pitch, Lt965R, Lumenera, Co.) for $(x, y, \lambda)$ imaging and a TDI camera (4640 × 256 resolution, 5 μm pixel pitch, VT-4K5X-H200, Vieworks, Co.) for $(x, y, t)$ imaging and $(x, y, \lambda, t)$ imaging. In order to block the beams with unwanted diffractions and DC noises, we fabricated a pupil mask with nine apertures (diameter = 2.5 mm each) and placed it at a proximity of the diffraction grating (at Fourier plane).

In the imaging flow cytometry demonstration, we flowed fluorescent beads (diameter = 15 μm, F8841 FluoSpheres, Thermo Fisher Scientific) in a custom microfluidic channel and controlled the flow rate using a syringe pump (NE-1010, New Era Pump Systems). The width, height, and length of the microfluidic channel are 140 μm, 25 μm, and 30 mm, respectively. The imaging FOV is approximately 400 μm × 140 μm. The scene was illuminated by a nanosecond pulsed laser ($\lambda = 532$ nm, FQS-200-1-532, Elforlight Ltd.) and captured by the TDI camera at 200 kHz.

**Image formation in direct optical mapping**. The relation between an incident 3D datacube, $I(x, y, z)$ and a measured 2D datacube, $E(u, v)$ through our system can be described as

$$E(u, \ v) = \mathbf{TSC}I(x, y, z),\tag{1}$$

where $\mathbf{C}$ is an operator describes the rearrangement of the spatial content of the incident scene by the active optical mapper. The $x$- and $y$-directions are horizontal and vertical directions in spatial axes. The $z$-direction is either spectral or temporal axis of the 3D datacube. $\mathbf{S}$ and $\mathbf{T}$ are linear shearing and integration operators, respectively. In the direct optical mapping method, the operator $\mathbf{C}$ is set to directly connect the voxels of the incident 3D datacube to the pixels of the 2D image sensor in a one-to-one manner: $E(i) = \mathbf{A}_{ij}I(j)$ where $\mathbf{A} = \mathbf{TSC}$ is an invertible reshaping operator. $i$ and $j$ are indices of voxels in the 2D $(u, v)$ image and 3D $(x, y, z)$ datacubes, respectively. Therefore, the 3D datacube can be reconstructed by scalar multiplications

$$I(j) = \mathbf{A}_{ij}^{-1}E(i) \text{where } \mathbf{A}_{ij}^{-1} = 1/\mathbf{A}_{ij}\left(\mathbf{A}_{ij} \neq 0\right).\tag{2}$$

**Phase pattern design for demultiplexing datacubes in direct optical mapping**. The optimal design of the phase pattern displayed by the SLM is crucial for mapping the incident 3D datacube onto the 2D detector in the one-to-one manner. Due to the pixelated nature of the SLM, the maximum achievable deflection angle

of the beam is given as $\theta_m = \sin^{-1}(\lambda/2p)$, where $\lambda$ and $p$ are the wavelength of the beam and the pixel pitch of the SLM, respectively. If the information of the incident image is carried beyond $\theta_m$, aliasing artefacts would occur and form ghost images. In addition, the SLM suffers from zero-order (un-diffracted light) and other noises, which arise from a finite fill factor and insufficient dynamic ranges. Also, the diffraction of the beam through relaying optics with a finite aperture should be accounted for when designing the phase pattern.

In our hyperspectral imaging implementation, the SLM slices the incident image into 45 segments along the vertical direction. The number of SLM pixels corresponding to each slice is 42 × 1152 $(x, y)$. The telecentric imaging system with an adjustable iris was used to match the size of the point spread function (PSF) of the incident image to the spatial resolution of the system defined as the width of the sliced image at the SLM plane. The 45 image slices were divided into nine groups. By displaying nine phase ramps corresponding to the centre positions of each lenslet on the array, the groups of sliced images propagated into nine directions. Specifically, with $\lambda = 550$ nm and $p = 9.2$ μm, the maximum deflection angle of the SLM, $\theta_m \approx 0.03$ rad. The propagating angles of the groups of images $(\theta_x, \theta_y)$ were equally distributed between 0 to $\theta_m$ in both $x$- and $y$-directions in order to avoid overlaps from both the zero-order (un-diffracted) noise and aliasing artefacts from higher-order diffractions; $\theta_x = (0.005, 0.015, 0.025)$ and $\theta_y = (0.005, 0.015, 0.025)$. After the beams were reflected from the SLM and passed through the achromatic lens ($f = 250$ mm), the spatial separations between each image slices at the focal distance were $f \times \Delta\theta = 2.5$ mm. Therefore, the lenslet array was designed to possess a 2.5 mm centre to centre distance between neighbouring lenses. The magnification from the SLM plane to the sensor plane was 0.12, and the image slice of width 386.4 μm (42 SLM pixels) was imaged onto the camera at a width of 46 μm size. The total size of the image incident on the camera was 7.5 mm × 7.5 mm, where the size of the PSF given by the backend optics, the lenslet array, was 8.4 μm. Therefore, the camera with a sensor size of 12.4 mm × 10.0 mm and a pixel pitch of 3.69 μm effectively sample the incident image. The resultant image was a set of vertically sliced images with the desired spacings in the horizontal direction. The spacings were filled with the spectral information by using the grating placed at the conjugated Fourier plane of the image. A band-pass filter corresponding to the spacing was used to limit the spectral range of the measurement to avoid the overlap of spectral information from different image slices.

Likewise, in high-speed imaging implementation, the incident scene was horizontally sliced into 27 segments. Only three lenses at the middle row of the 3 × 3 lenslet array were used to image the scene. Therefore, nine image slices share a single lens in the lenslet array. The vertical spacings between the sliced images were filled with the temporal information by the TDI operation.

**TDI for high-speed imaging**. We employed a TDI camera as a temporal shearing and integration device. In TDI operation, the series of the images, $I(x, y, t)$, is captured by the 2D sensor array in the camera at a given frame rate. The resultant photoelectrons from each time frame are transferred along the vertical axis of the camera as a function of time. During measurement, the charge packets are successively accumulated and transferred until being read out by the horizontal register. The output signal is a series of vertically integrated horizontal lines. In direct optical mapping with a uniform magnification, the total photoelectrons contained at $m$th pixel in the $l$th line image can be described as, $E(m, l) = I(m, l, l/v + a)$ where $v$ and $a$ are scalar constants describe a temporal shearing speed and a vertical (temporal) offset of TDI stages for a given measurement time window, respectively. We note that the operation of the TDI camera can be considered as a time-domain counterpart of spectral shearing and integration using a diffraction grating and an ordinary time-integrated 2D camera. The TDI camera linearly shears the image as a function of time, whereas the diffraction grating linearly shears the image as a function of the spectral frequency.

In our demonstration, the transient scene was horizontally sliced into 27 segments. The image segments were captured by 2048 × 256 $(x, y)$ pixels in the sensor of the TDI camera. The scene was transferred and integrated along the vertical direction of the camera. While running the TDI camera at full 256 TDI stages, an optical chopper was used to confine the time window of the measurement. The effective number of TDI stages given by the time window was 28, avoiding the overlap of signals from adjacent horizontal slices in the vertical direction. The time-series of horizontal line data of 2048 pixels were transferred to the host computer for the reconstruction of high-dimensional datacubes.

**System calibration for direct optical mapping**. To account for the aberrations, misalignment of optics, and non-uniform spectral sensitivity of the sensor, we experimentally calibrated the mapping relations between voxels of the 3D datacube $I(x, y, z)$ and pixels of the image sensor $E(u, v)$.

First, we calibrated mapping relations between spatial coordinates of the 3D datacube and the 2D pixels of the image sensor at a single frequency ($\lambda = 532$ nm). The system was illuminated by a collimated laser beam where a fast-rotating diffuser was added to eliminate speckle noises. The resultant images were a set of line images which we called 'elemental images'. The positions of the elemental images on the sensor $E(u, v)$ corresponding to the voxels of the incident image $I(x, y, z = 532$ nm) were mapped. During the mapping process, each elemental line image was fitted by independent curve to compensate for lens-dependent image distortions and the misalignment between the SLM and the sensor.

Next, we calibrated the spectral response of the elemental images in order to find the complete mapping relations for hyperspectral imaging. We note that the spectral dispersion direction of the elemental images was precisely matched to the horizontal direction of the camera pixels. The system was illuminated by collimated beams with five different wavelengths which were generated by putting narrow-band spectral filters ($\lambda = 510, 532, 550$, and $590$ nm) in front of a spatially filtered halogen lamp beam. The linear spectral response curve was found by fitting the data points which relate the wavelengths of the beams and horizontal locations of the pixels that read maximum intensity. With linear interpolations, a complete lookup table that relates the 3D datacube $I(x, y, z)$ and the 2D sensor $E(u, v)$ was obtained. For high-speed imaging, the temporal shearing speed (pixels per seconds) was precisely controlled by the internal clock of the TDI camera.

The diffraction efficiency of the SLM varies as a function of impinging spatial frequencies on it. To compensate for this artefact, we illuminated the system with a uniform plane beam, which was generated by expanding a beam from the halogen lamp with a band-pass filter (CWL = 550 nm, FWHM = 40 nm). The acquired image was used to normalise the non-homogeneous diffraction efficiency of the SLM. In addition, a spectral quantum efficiency curve of the image sensor was used to normalise the spectral sensitivity.

**Image formation in compressed measurement**. For the compressed sensing method with nine complementary encoding patterns, the encoding operator $\mathbf{C}$ can be described as $\mathbf{C} = [\mathbf{C}_1, \mathbf{C}_2, \dots \mathbf{C}_9]^T$ where $\mathbf{C}_i$ ($i = 1, 2, \dots, 9$) is the $i$th impinging pattern with $\mathbf{C}_1 + \mathbf{C}_2 + \dots + \mathbf{C}_9 = \mathbf{I}$. The sensor captures nine temporally sheared and integrated images, $E_i$ ($i = 1, 2, \dots, 9$), simultaneously

$$[E_1, E_2, \dots, E_9]^T = \mathbf{TS}[\mathbf{C}_1, \mathbf{C}_2, \dots, \mathbf{C}_9]^T I. \quad (3)$$

Given a unit magnification and an ideal imaging system, the image formed at the SLM plane, $I(x', y', z)$, is identical to the incident image, $I(x, y, z)$. The encoded image at the SLM plane can be described by the following equation:

$$I_{C_i}(x', y', z) = I(x', y', z)C_{i,j,k}\mathrm{rect}\left[\frac{x'}{d'} - \left(j + \frac{1}{2}\right), \frac{y'}{d'} - \left(k + \frac{1}{2}\right)\right] \text{ with } i = 1, 2, \dots, 9. \quad (4)$$

Here, $C_{i,j,k}$ is an element (0 or 1) of the matrix representing the binary spatial pattern impinged on the image. $j$ and $k$ are matrix element indices. $\mathrm{rect}(X)$ is the rectangular function defined as $\mathrm{rect}(X) = 1$, if $|x| < 1/2$ and 0 else. $d'$ is the minimum feature size of the pseudorandom pattern impinged by the SLM. Assume that the shearing operator shears the datacube linearly along the vertical axis of the sensor ($y''$), the resultant image at the sensor is given as, $I_{S_i}(x'', y'', z) = I_{C_i}(x'', y'' - vz, z)$ where $v$ is the shearing velocity. The photoelectrons captured at the sensor plane can be discretised by using the sensor's pixel pitch, $d''$ as shown below.

$$E_{C_i}(m, n, k) = \eta \int dx'' dy'' dz I_{C_i}(x', y', z)$$
$$\mathrm{rect}\left[\frac{x''}{d''} - \left(m + \frac{1}{2}\right)\right]\mathrm{rect}\left[\frac{x''}{d''} - \left(n + \frac{1}{2}\right)\right]\mathrm{rect}\left[\frac{z}{\Delta_z} - \left(k + \frac{1}{2}\right)\right], \quad (5)$$

where $\eta$ is the sensor's quantum efficiency and $\Delta_z = d''/v$.

In TDI operation, the charge packet is transferred along the vertical axis of the sensor ($y''$) and continuously accumulates photoelectrons. We assume that the sensor's pixel pitch is identical to the minimum feature size of the pseudo-random pattern ($d'' = d'$). Upon readout, the total photoelectrons contained at $m$-th pixel in the $l$-th line image output from the channels are given as

$$E_{C_i}^*(m, l) = \sum_{p=1}^{P} E_{C_i}(m, p, p + l - 1) = \eta \frac{d''^3}{v} \sum_{r=l+1}^{l+P} C_{i,m,r-l} I_{m,r-l,r}, \quad (6)$$

where $P$ is TDI stages which is the number of the pixels in the TDI camera along the direction of integration. The same model also applies to the hyperspectral imaging scheme. In this case, $P$ becomes the spectral bandwidth of the measurement in terms of pixel numbers along the dispersion axis, and $v$ becomes the shearing velocity along the spectral frequency axis.

The 3D datacube $I(x, y, z)$ can be reconstructed by solving the associated inverse problem. In this work, we solved the minimisation problem:

$$\mathrm{argmin}\left\{\frac{1}{2}\left\|\left[E_{C_1}^*, E_{C_2}^*, \dots, E_{C_9}^*\right]^T - \mathbf{TS}[\mathbf{C}_1, \mathbf{C}_2, \dots, \mathbf{C}_9]^T I\right\|_2^2 + \lambda\Phi(I)\right\}, \quad (7)$$

where $\lambda$ and $\Phi(I)$ are a regularisation parameter and a regularisation function, respectively. To solve the problem, we adopted a total variation regularizer with BM3D denoiser[40] and a two-step iterative shrinkage/thresholding algorithm[32].

**Calibration for compressed sensing**. We experimentally measured the encoding operator $\mathbf{C}$ as it can differ from our designed pseudo-random pattern due to experimental artefacts such as aberrations from imperfect optics, finite PSF of the system, nonuniform diffraction efficiency of the SLM, and beams from undesired diffracted orders. To record the encoding operator, we illuminated the system with a collimated laser beam ($\lambda = 532$ nm). A fast-rotating ground glass diffuser was used to eliminate speckle noises. No temporal shearing was applied during the measurement. Therefore, the encoding pattern was achieved simply by measuring the intensity value at the sensor. In practice, the acquired encoding operator was not a binary pattern but a spatially varying intensity map. We measured the operator with a high dynamic range by combining multiple images captured at varying exposures. By using the measured intensity map instead of the designed binary pattern, we took account of the experimental artefacts.

**Reporting summary**. Further information on research design is available in the Nature Research Reporting Summary linked to this article.

## Data availability

The data that support the plots within this paper and other findings of this study are available from the corresponding author upon reasonable request.

## Code availability

Codes used for this work are available from the corresponding author upon reasonable request.

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

## Acknowledgements
We thank Congnyu Che and Brian T. Cunningham for providing custom microfluidic chips. We also thank Mantas Zurauskas, Rishyashring R. Iyer, and Stephen A. Boppart for helpful discussions. This work was supported partially by National Institutes of Health (R01EY029397, R35GM128761, and R21EB028375); National Science Foundation (1652150). J.P. acknowledges partial support from Basic Science Research Programme through the National Research Foundation of Korea (NRF) funded by the Ministry of Education (2019R1A6A3A03031505).

## Author contributions
J.P. and L.G. designed the system. J.P. built the system, performed the experiments, and analysed the data. X.F. and R.L. contributed analytic tools. J.P. and L.G. wrote the paper with input from all authors. L.G. supervised the project.

## Competing interests
The authors declare no competing interests.
