## [Peer Review File · Nature Communications]

REVIEWER COMMENTS

Reviewer #1 (Remarks to the Author):

This paper proposes a programmable focal plane coding system for high speed and hyperspectral imaging. Construction of such a system is a good idea, although 4F design makes such systems relatively large, especially in the usually limited field of view accepted by spatial light modulators. The paper would be stronger if design trade-offs and system performance (SNR, computational time, sensitivity) were compared for various operating modes and against previous similar systems.

The main claim of the paper is that dynamic coding enables a flexible pixel sampling function for diverse applications in spectral and temporal imaging. The combination of an SLM, hyperspectral imaging and high frame rate imaging may be new, but each of these aspects has been previously explored. See, for example, Tsung-Han Tsai, Patrick Lull, Xin Yuan, Lawrence Carin, and David J. Brady, "Spectral-temporal compressive imaging," *Opt. Lett.* 40, 4054-4057 (2015) for a system capturing spectral and high frame rate data at the same time. Alternatively Matthew Dunlop-Gray, Phillip K. Poon, Dathon Golish, Esteban Vera, and Michael E. Gehm, "Experimental demonstration of an adaptive architecture for direct spectral imaging classification," *Opt. Express* 24, 18307-18321 (2016) presents a dynamic coding as proposed here. Similarly, D. Reddy, A. Veeraraghavan, and R. Chellappa, "P2c2: Programmable pixel compressive camera for high speed imaging," in *CVPR 2011*, June 2011, pp. 329–336. and Y. Hitomi, J. Gu, M. Gupta, T. Mitsunaga, and S. K. Nayar, "Video from a single coded exposure photograph using a learned over-complete dictionary," in *2011 International Conference on Computer Vision*, Nov 2011, pp. 287–294. used SLM modulators for temporal encoding. While there are some new wrinkles in the optical design here, I don't see enough novelty to justify publication in *Nature Communications*. I am sure, however, that the authors can use the impressive system they have constructed to generate publishable results. A good strategy may be to compare some aspects of optical or computational performance to explain how this system pushes the state of the art forward.

Reviewer #2 (Remarks to the Author):

The manuscript by Park et al reports a flexible capturing device for multidimensional photography. An SLM is used to deflect the light beam with different angles. As such, it functions as an array of angled mirror facets, slicing the image into strips and creating blank spaces in between. High-dimensional datacube such as spectrum and temporal information can be dispersed in the blank spaces for 2D single-shot acquisition. The reported system presents an advantage in switching between direct and compressed measurements. It also has the advantage of performing various tasks ranging from hyperspectral to ultrafast imaging in a single device upon demand. The authors have presented several application modes using the system, including both direct and compressive imaging on 2D + spectrum and 2D + time, and lastly, 2D + spectrum + time. The presented data are solid and technically sound. The results on 2D + spectrum + time are impressive and appealing. In my opinion, this manuscript is in high quality and will generate impacts for different research communities such as spectroscopy, microscopy, computational optics, and biophotonics. I have the following comments that the authors may consider to better address.

1. Even for the compressive measurement, there is still a relatively large gap between the 9-million camera pixels and the voxel number of multi-dimensional measurements. It seems that it is also limited by the pixel and phase range on SLM. Some discussions are needed on how to better close this gap.

2. The reported device is still limited by the throughput of a single camera. It essentially tradeoffs the field of view for the other dimensions of the datacube. Perhaps, a more appealing strategy is to use multiple cameras. I understand there is a constraint on the deflection angle given by the SLM. Maybe a longer focal length can be used in that case? I suggest the authors comment on the aspect of employing multi cameras for the reported system.

We thank both reviewers and the editor for their insightful comments, which have greatly improved the quality of our work. Below we provide point-to-point responses. The original referee comments are provided in black, whereas our replies are given in blue. Contents in the revised manuscript addressing the reviewers' concerns are given in red.

Reviewer #1 (Remarks to the Author):

This paper proposes a programmable focal plane coding system for high speed and hyperspectral imaging. Construction of such a system is a good idea, although 4F design makes such systems relatively large, especially in the usually limited field of view accepted by spatial light modulators. The paper would be stronger if design trade-offs and system performance (SNR, computational time, sensitivity) were compared for various operating modes and against previous similar systems.

The main claim of the paper is that dynamic coding enables a flexible pixel sampling function for diverse applications in spectral and temporal imaging. The combination of an SLM, hyperspectral imaging and high frame rate imaging may be new, but each of these aspects has been previously explored. See, for example, Tsung-Han Tsai, Patrick Lull, Xin Yuan, Lawrence Carin, and David J. Brady, "Spectral-temporal compressive imaging," *Opt. Lett.* 40, 4054-4057 (2015) for a system capturing spectral and high frame rate data at the same time. Alternatively Matthew Dunlop-Gray, Phillip K. Poon, Dathon Golish, Esteban Vera, and Michael E. Gehm, "Experimental demonstration of an adaptive architecture for direct spectral imaging classification," *Opt. Express* 24, 18307-18321 (2016) presents a dynamic coding as proposed here. Similarly, D. Reddy, A. Veeraraghavan, and R. Chellappa, "P2c2: Programmable pixel compressive camera for high speed imaging," in *CVPR 2011*, June 2011, pp. 329–336. and Y. Hitomi, J. Gu, M. Gupta, T. Mitsunaga, and S. K. Nayar, "Video from a single coded exposure photograph using a learned over-complete dictionary," in *2011 International Conference on Computer Vision*, Nov 2011, pp. 287–294. used SLM modulators for temporal encoding. While there are some new wrinkles in the optical design here, I don't see enough novelty to justify publication in *Nature Communications*. I am sure, however, that the authors can use the impressive system they have constructed to generate publishable results. A good strategy may be to compare some aspects of optical or computational performance to explain how this system pushes the state of the art forward.

We appreciate the reviewer for careful reading and helpful suggestions. We agree with the reviewer that our snapshot multidimensional photography method share similar roots as mentioned approaches in parts. However, as an integrated device, our system is highly innovative in two aspects:

- First, our method is based on **active optical mapping** while all previous systems are based on passive mapping. The ability to manipulate the mapping relation between light datacube voxels and camera pixels endows great flexibility in measurement, allowing the system to tailor an application and readily switch between direct and compressed imaging. To our knowledge, this is the first time such **a transformable system** has been demonstrated in high-dimensional optical imaging, creating a new category of imagers that have been long sought-after. To spotlight this advantage, we added a new **Supplementary Table 1** comparing our method with previous systems in three key specs:

	Active-mapping snapshot multidimensional photography	IMS ¹	CASSI ²	AFSS ³	P2C2 ⁴	CACTI ⁵	CUP ⁶	STAMP ¹⁰	Coded exposure photograph ¹¹
Optical mapping mode	Active	Passive	Passive	Passive	Passive	Passive	Passive	Passive	Passive
Light datacube measured	(x, y, λ) , (x, y, t) , or (x, y, λ, t)	(x, y, λ)	(x, y, λ)	(x, y, λ)	(x, y, t)	(x, y, t)	(x, y, t)	(x, y, t)	(x, y, t)
Operating mode	Direct, compressed, or hybrid	Direct	Compressed	Compressed	Compressed	Compressed	Compressed	Direct	Compressed

Supplementary Table 1. Comparison of active-mapping snapshot multidimensional photography against previous passive systems. IMS, image mapping spectrometer; CASSI, coded aperture snapshot spectral imager; AFSS, Adaptive Feature-Specific Spectrometer; P2C2, programmable pixel compressive camera; CACTI, coded aperture compressive temporal imaging; CUP, compressive ultrafast photography; STAMP, sequentially timed all-optical mapping photography.

- Second, our system consists of many innovative elements that significantly improve the imaging performance and enable applications previously not possible. For example, for the first time, we showed that, by using a cylindrical mapping pattern, we can increase the spectral resolution by a factor of two, compared with its state-of-the-art passive mapping counterpart (Image Mapping Spectrometer) (Supplementary Note 1). As another example, we demonstrated high-speed imaging using a time-delay integration (TDI) camera. As discussed in **Methods: Time-delay integration for high-speed imaging**, TDI cameras have been conventionally used as a line imager. We, again for the first time, showed that it can be employed as a temporally shearing device, thereby allowing 2D ultrafast imaging at the camera's line readout speed and without moving parts. This novel use of TDI cameras itself can inspire many interesting applications in high-speed optical imaging. To demonstrate it, we added a new experimental result (**Fig.6e-f**) showing a proof-of-concept application in imaging flow cytometry.

Fig. 6e-f. Experimental demonstration of active-mapping snapshot multidimensional photography in high throughput imaging flow cytometry. **e**, Photograph of a microfluidic channel. **f**, Representative temporal frames of a flowing fluorescent bead imaged at 200 kHz.

We also added the experimental description in the main text section “(x, y, t) imaging with compressive sensing” that reads:

“We further demonstrated a niche application of our system in high-throughput imaging flow cytometry^{35–37}. We flowed a fluorescent bead (diameter, 15 μm) in a custom microfluidic channel at a velocity of 0.8 m/s using a syringe pump and illuminated the scene with a nanosecond pulsed laser ($\lambda = 532 \text{ nm}$) (Fig. 6e). The emitted fluorescence was collected by an objective lens ($\times 4/0.16\text{NA}$), filtered by a dichroic mirror, and imaged by our system at 200 kHz. The reconstructed blur-free fluorescent bead images at representative temporal frames are shown in Fig. 6f.”

Below we address the reviewers’ specific concerns:

- System design constraint

We added new discussions about the system design constraint of using a pixelated SLM in **Discussion: Tunable resolution** (Main text, page 9) and **Supplementary Note 2**. In brief, the measurable number of datacube voxels in a snapshot is fundamentally limited by the pixel number of the SLM. Although the current work mainly focuses on snapshot measurements, we added new results and provided a strategy to break this limitation with multiple measurements (**Supplementary Figure 6**) by actively tuning the mapping relations between multidimensional datacube voxels and 2D camera pixels.

We added the following paragraph in **Discussion: Tunable resolution**:

“In direct measurement, the product of the spatial and spectral/temporal sampling rates (i.e., the total number of voxels in the light datacube) is fundamentally limited by: (1) the number of controllable degrees of freedom of the pixelated SLM, which is referred to as the space-bandwidth product (SBP) of the system (Supplementary Note 2)³⁷, and (2) the total number of pixels of the image sensor. Because the pixel counts of large-format image sensors far exceed that of current state-of-the-art SLMs, the performance of our system is bounded by the SBP of the SLM. Yet, the use of a large-format image sensor or even an array of image sensors^{28,38,39} is beneficial to image reconstruction, especially for a highly compressed measurement scheme.”

The new section **Supplementary Note 2** reads:

“Supplementary Note 2. Design constraint due to the limited space-bandwidth product of an SLM

Although multidimensional photography with an active optical mapper provides flexibility on measurement, achievable spatial and spectral/temporal sampling rates (the number of voxels of a light datacube) in the direct measurement are fundamentally limited by the number of controllable degrees of freedom (i.e., pixel counts) of the SLM, which is also referred to as space-bandwidth product (SBP) of the SLM.

Given an SLM with (n_x, n_y) pixels and a pixel pitch of p , an incident image is sliced and reflected by the SLM towards desired angles. To fully utilise the SBP of the SLM, we must choose angles that range across the maximum diffraction angle of the SLM, $\theta = \frac{\lambda}{p}$. Sliced images propagate through a lens with a focal length of f_1 . A $l \times l$ microlens array captures the full Fourier spectrum of the SLM. Therefore, the relation between the diameter, d , of the each microlens and the number of microlenses is given as $d = f_1 \frac{\lambda 1}{p l}$. Given m SLM pixels spanning the width of a mirror slice and k mirror slices imaged through the same microlens, the number of microlenses needed can be computed as $l \times l = \frac{n_x 1}{m k}$.

The spatial and spectral/temporal sampling rates of the light datacube are given as $N_x = \frac{n_x}{m}$, $N_y = \frac{n_y}{m}$, and $N_{\lambda,t} = \frac{n_x 1}{m k}$, respectively. The total number of voxels in the light datacubes is:

$$N_x \times N_y \times N_{\lambda,t} = n_x n_y \frac{n_x}{m^3 k} = N_{\text{SLM}} \frac{n_x}{m^3 k},$$

where N_{SLM} is the SBP of the SLM. The microlens' point-spread-function (PSF) is given by:

$$PSF = C \frac{f_2 \lambda}{d/2}, \quad C = 1.22 \text{ for Rayleigh criterion.}$$

Here f_2 is the focal length of the microlenses.

In our system, we match the mirror slice width at the camera plane, $w = mp \frac{f_2}{f_1}$, with the microlens' PSF:

$$mp \frac{f_2}{f_1} = C \frac{f_2 \lambda}{d/2},$$

Substituting d with $f_1 \frac{\lambda 1}{p l}$ gives

$$m = 2Cl.$$

Further substituting l with $\sqrt{\frac{n_x 1}{m k}}$ leads to

$$\frac{n_x}{m^3 k} = \frac{1}{4C^2}.$$

Finally, we can then rewrite the total number of light field voxels as:

$$N_x \times N_y \times N_{\lambda,t} = N_{\text{SLM}} \frac{n_x}{m^3 k} = \frac{1}{4C^2} N_{\text{SLM}}.$$

Therefore, the voxel number of the light datacube is limited by the SBP of the SLM. To break this limitation, we can adopt a temporal-multiplexing strategy using multiple measurements. We show an example in Supplementary Figure 6.”

Supplementary Figure 6. Spatial resolution enhancement in direct optical mapping with multiple measurements. A 1951-USAF resolution target was illuminated with an incoherent beam (Central wavelength = 550 nm, Full-width at half maximum = 10 nm) and served as an object. **a**, Snapshot measurement. The width of vertical mirror slices is 42 SLM pixels. **b**, Multiple measurements. The SLM pattern was divided into nine sub-patterns with equal spacing. Therefore, the width of slices in sub-patterns is 14 SLM pixels. A sub-pattern was displayed at a time, and a total of nine images were captured sequentially. Features of the incident image smaller than the width of the vertical slices in the snapshot measurement scheme (42 SLM pixels) were resolved. **c**, Image captured with snapshot measurement at 550 nm. **d**, Image captured with multiple measurements at 550 nm. **e**, Ground-truth image captured with a monochromatic camera. **f**, Intensity profiles of the images in (c) and (d) along red dotted lines. The image from multiple measurements shows higher contrast with three times higher spatial sampling rates along the horizontal axis.

- System performance comparison among different operating modes

The revised manuscript includes a characterization of the system when operating in direct and compressed measurement modes. For direct measurement, we imaged a standard color checker target and quantified the spectral accuracy. As shown in Fig. 8, an overall normalised root-mean-squared-error (RMSE) was measured as 0.107, representing a high-fidelity measurement. Based on the same hardware, compressed measurement exhibits the same optical performance as that in direct measurement. However, the quality of the image reconstructed is sensitive to the SNR and compression ratio. To show this dependence, we performed a simulation and plot the peak-signal-to-noise ratio and image correlation coefficient as functions of the SNR and compression ratio in Fig. 9c. The result implies that, with a high SNR and a low compression ratio, the image quality of compressed measurement approaches that in direct measurement.

Additionally, the reconstruction of a 3D datacube of size $400 \times 340 \times 275$ (x, y, λ) using a compressed measurement scheme requires approximately 20 minutes on a personal computer. In contrast, direct optical mapping uses a lookup table to connect measurement to 3D datacube voxels with negligible computational time. Therefore, direct measurement is superior to compressed measurement in computational cost.

In **Discussion**, we added a new section “**System evaluation in direct measurement mode**” that reads:

“We quantitatively evaluated the system performance in direct measurement mode. Without loss of generality, we assessed the system in spectral imaging of a standard colour checker target. We illuminated the colour checker with a green light (CWL = 550 nm, FWHM = 40 nm) and captured spectrum of the scattered light using both our system and a benchmark fibre spectrometer (Figure 8). Here we use the fibre spectrometer to provide the ground-truth measurement. The root means squared errors (RMSE) of the normalised spectrum were

quantitatively analysed (Fig. 8d). The average value of the RMSE was 0.107. Noises are mainly contributed by the stripe artefacts caused by non-uniform light diffraction efficiency of the SLM and unwanted diffraction orders of the beam. In addition, with a lower illumination irradiance, the RMSE tends to be higher because the signal-to-noise ratio (SNR) is relatively low.”

Fig. 8 Technical evaluation of (x, y, λ) imaging with direct optical mapping. **a**, Photograph of the colour checker captured by a colour camera when it was illuminated by a fluorescent lamp. **b**, Photograph of the colour checker captured by a monochromatic camera when it was illuminated by the green light. **c**, Normalised spectral irradiances of the light scattered from the colour checker measured by multidimensional photography and a commercial fibre. The graphs indicate that our method can capture the hyperspectral datacube with high fidelity. **d**, Root-mean-squared error (RMSE) of the hyperspectral images captured by multidimensional photography provided that the spectrum from the fibre spectrometer gives the ground-truth values. The averaged value of RMSE was 0.107.

In **Discussion**, we also added a new section “**Direct measurement versus compressed measurement**” that reads:

Our system presents a prominent advantage in flexible switching between direct and compressed measurements, both of which have pros and cons. On the one hand, while the direct mapping always yields accurate measurement due to the invertibility of unique optical mapping relation, it faces a trade-off between information content along myriad dimensions for a given detector array with a limited pixel number. On the other hand, the compressed measurement allows capturing a large-sized datacube beyond the Nyquist sampling rate of the system. However, the reconstruction process is computationally extensive, and it is prone to generating artefacts. For example, the reconstruction of a light datacube of size $400 \times 340 \times 275$ (x, y, λ or t) takes approximately twenty minutes on a personal computer (i7-8700 CPU, 4.6 GHz clock rate).

To analyse at which conditions the compressed measurement scheme prevails over its direct counterpart, we evaluated the noise tolerance using a numerical approach. We generated a 3D ($150 \times 100 \times 50$) (x, y, λ or t) datacube with varied SNR as the input scene, and we simulated three image formation models by encoding the scene with a binary pattern, three complementary patterns, and nine complementary patterns, respectively. The datacube was sheared and integrated along the 3rd axis (spectrum or time). The resultant 2D scene was recorded by a 2D (150×100) detector with white Gaussian noise corresponding to the given SNR. We defined the compression ratio, ξ , as the ratio of the voxel number of the incident 3D datacube to the pixel number of the 2D detector. For example, if the 3D ($150 \times 100 \times 50$) scene is encoded by a single binary pattern and one 2D (150×100) image is captured, $\xi = 50$. If the scene is divided and captured through nine complementary encoding masks simultaneously ($150 \times 100 \times 9$), $\xi = 50/9$. If the 3D datacube is directly mapped onto a 2D sensor with a large number of pixels on a one-to-one basis, $\xi = 1$. The reconstruction results were quantified based on two metrics: the correlation coefficient and peak signal-to-noise ratio. Figure 9 shows the reconstruction fidelity of the 3D datacube. When the SNR is 25 or higher, we can observe the expected scene in all simulations. However, the fine feature is visible only when the 3D datacube is captured with nine complementary encoding patterns ($\xi = 50/9$). In this case, the correlation coefficient between the noise-free 3D datacube reaches over 0.95, a level that is comparable to the direct mapping ($\xi = 1$) with the denoising operations⁴¹. Because the reconstruction fidelity highly depends on the sparsity of the signal, we cannot quantitatively compare the simulation with experiments. Nonetheless, the results herein exhibit the same trend as our previous experimental results—the hyperspectral imaging with nine complementary encoding patterns shows fine features (Fig. 4) while the ultrafast imaging with three complementary encoding patterns results in blurred edges (Fig. 6).

Fig. 9. Effect of the compression ratio on compressed sensing. **a**, 3D datacube for the simulation. A total of 50 frames are compressed into a single frame. The 21st to 30th frames contain the image of the dice, and the rest 40 frames are blank. **b**, Reconstructed 3D datacube. The 25th frames at given compression ratios and signal to noise ratios are shown. **c**, Peak signal-to-noise ratio and correlation coefficient of the reconstructed datacube given the noise-free solution in (a).

Reviewer #2 (Remarks to the Author):

The manuscript by Park et al reports a flexible capturing device for multidimensional photography. An SLM is used to deflect the light beam with different angles. As such, it functions as an array of angled mirror facets, slicing the image into strips and creating blank spaces in between. High-dimensional datacube such as spectrum and temporal information can be dispersed in the blank spaces for 2D single-shot acquisition. The reported system presents an advantage in switching between direct and compressed measurements. It also has the advantage of performing various tasks ranging from hyperspectral to ultrafast imaging in a single device upon demand. The authors have presented several application modes using the system, including both direct and compressive imaging on 2D + spectrum and 2D + time, and lastly, 2D + spectrum + time. The presented data are solid and technically sound. The results on 2D + spectrum + time are impressive and appealing. In my opinion, this manuscript is in high quality and will generate impacts for different research communities such as spectroscopy, microscopy, computational optics, and biophotonics. I have the following comments that the authors may consider to better address.

We thank the reviewer for the overall comments on our work and the kind compliment. The reviewer's concerns have been addressed in the following.

1. Even for the compressive measurement, there is still a relatively large gap between the 9-million camera pixels and the voxel number of multi-dimensional measurements. It seems that it is also limited by the pixel and phase range on SLM. Some discussions are needed on how to better close this gap.

We thank the reviewer for the helpful suggestion. As the reviewer noted, the large gap between the camera pixel count and the datacube voxel number originates from the limited pixel number of the SLM. We discussed the optical design strategy on how to fully utilise the pixelated SLM in the revised manuscript (Supplementary Note 2). Also, we derived a relation between the pixel number of the SLM and the achievable voxel number in the system. In short, the maximum achievable voxel number in the snapshot measurement scheme is fundamentally limited by the pixel number (space-bandwidth product) of the SLM.

In practice, the pixel counts of modern image sensors surpass that of current state-of-the-art SLMs. In the revised manuscript, we provide a strategy on how this gap can be mitigated with multiple measurements by exploiting the *active* nature of the optical mapper (Supplementary Figure 6). Please see our detailed responses to Reviewer 1's comments. We would like to note that the versatility of the active optical mapper enables various measurement schemes, not limited to snapshot measurements demonstrated in this manuscript.

2. The reported device is still limited by the throughput of a single camera. It essentially tradeoffs the field of view for the other dimensions of the datacube. Perhaps, a more appealing strategy is to use multiple cameras. I understand there is a constraint on the deflection angle given by the SLM. Maybe a longer focal length can be used in that case? I suggest the authors comment on the aspect of employing multi cameras for the reported system.

As the reviewer noted, the pixel counts of both the SLM and camera are limiting factors. With current advances in digital image sensors, the throughput of our multidimensional photography is bounded by the limited pixel count (information capacity or space-bandwidth product) of the SLM; the maximum pixel count of a typical commercial SLM is about 8.3 million (3840×2160 resolution), whereas 100-megapixel cameras are widely available even in mobile devices. We mapped 3D datacube voxels to 2D camera sensor pixels through the multiscale $4f$ imaging systems. Due to its *imaging nature*, using lenses with different angular/lateral magnifications may change the size and spatial frequency extend (or effective diffraction angle) of the image, however, does not enhance the maximum information capacity of the system. In summary, the use of lenses with different focal lengths cannot change the throughput (in terms of information capacity) because pixel counts of available SLMs are far smaller than that of modern image sensors.

Although the current limitation is imposed by the pixel count of the SLM in the direct optical mapping scheme, the use of a large-format image sensor is beneficial to image reconstruction fidelity, especially for a highly

compressed measurement scheme. In the revised manuscript, we comment on this pixel-count-bottleneck and aspect of employing multiple cameras on page 9 of main text (tunable resolution section).

REVIEWERS' COMMENTS

Reviewer #1 (Remarks to the Author):

The authors have responded the original reviewer comments with a thorough and rigorous set of additions to the manuscript. This is a very solid contribution to the design of adaptable computational cameras, it should now be accepted for publication.

Reviewer #2 (Remarks to the Author):

The authors have well addressed my comments in the revised manuscript.

Reviewer #1 (Remarks to the Author):

The authors have responded the original reviewer comments with a thorough and rigorous set of additions to the manuscript. This is a very solid contribution to the design of adaptable computational cameras, it should now be accepted for publication.

Reviewer #2 (Remarks to the Author):

The authors have well addressed my comments in the revised manuscript.

We thank both reviewers for their insightful comments, which have greatly improved the quality of our work.